# Denosumab-Induced Immune Hepatitis

**DOI:** 10.3390/biomedicines9010076

**Published:** 2021-01-14

**Authors:** Viviana Ostrovsky, Stephen Malnick, Shahar Ish-Shalom, Nadya Ziv Sokolowskaia, Ady Yosepovich, Manuela Neuman

**Affiliations:** 1Diabetes, Endocrinology and Metabolic Disease Institute, Kaplan Medical Center, The Faculty of Medicine, The Hebrew University of Jerusalem, Rehovot 76100, Israel; viviana.ostrovsky@clalit.org.il; 2Department of Internal Medicine C. Kaplan Medical Center, The Faculty of Medicine, The Hebrew University of Jerusalem, Rehovot 76100, Israel; steve@stevemalnickmd.com; 3Department of Pathology Kaplan Medical Center, The Faculty of Medicine, The Hebrew University of Jerusalem, Rehovot 76100, Israel; shaharis@clalit.org.il (S.I.-S.); nadyaziv@clalit.org.il (N.Z.S.); adyyo@clalit.org.il (A.Y.); 4In Vitro Drug Safety and Biotechnology and Department of Pharmacology and Toxicology, University of Toronto, Toronto, ON M5G 1L5, Canada

**Keywords:** immune hepatitis, cytokine, denosumab, drug-induced hepatotoxicity, necrosis, nuclear factor-κB (NFκB), osteoporosis, receptor activator of nuclear factor-κB ligand (RANKL)

## Abstract

Denosumab–Prolia®, Xgeva® (Amgen) is a fully human antibody to the receptor activator of the nuclear factor-K ligand (RANKL). Hepatotoxicity is extremely rare, with only one reported case of immune origin. We present a second case of hepatotoxicity resulting from an immune reaction to denosumab. A 43-year-old female was referred to the Endocrinology, Diabetes & Metabolism Department for treatment of low bone mineral density (BMD) following endocrine therapy with letrozole and lucrin because of breast cancer. She developed premature menopause at the age of 36 years when she underwent a left lumpectomy due to an infiltrating duct carcinoma of the breast (T1 NO MO) and was subsequently started on endocrine therapy. Denosumab was started to prevent osteoporosis. On the third year after starting on denosumab and one month after she received the last injection, she became ill. The routine biochemical analysis showed that the levels of alanine aminotransferase (ALT) and aspartate aminotransferase (AST) rose appreciatively to 10 times the upper limit of normal (ULN). The gamma-glutamyl transferase (GGT) level was elevated slightly to 67 U/L (0–38 U/L). The serum gamma-globulin level was elevated to 1.72 g/dL (0.7–1.6 gr/dl), while the total bilirubin (TB) and serum albumin levels were normal. A liver biopsy revealed a moderate to severe chronic inflammatory infiltrate containing MUM-1 positive plasma cells. In addition, numerous CD-3 positive small T lymphocytes and few CD-20 positive B lymphocytes and eosinophils were seen in the portal tracts. Moderate to severe interface hepatitis, bile duct proliferation and mild portal fibrosis were also identified. The results could be consistent with the diagnosis of drug-induced liver injury (DILI).

## 1. Introduction

Denosumab is a human monoclonal antibody targeting the key bone resorption mediator receptor activator of the nuclear factor kappa B (NF-κB) ligand (RANKL). Additionally, denosumab contains odanacatib, a specific inhibitor of the osteoclast protease cathepsin K (OPC-K), and antibodies against the two endogenous inhibitors of bone formation, the proteins sclerostin and dickkopf-1.

The drug is administered via subcutaneous injection once every six months and is approved for the treatment of postmenopausal women with osteoporosis at an increased risk of fracture [1]. Denosumab is also one of the first-line treatments recommended for aromatase inhibitor-induced bone loss [2,3]. 

The onset of menopause is associated with a decrease of the estrogen concentration. The function of the receptor activator of the NF-κB ligand, RANKL, is to support the binding of the receptor activator RANK to the osteoclast [4].

Denosumab binding to RANKL inhibits osteoclast formation, maintenance and survival, and reduces bone resorption and turnover. It has a half-life of 25.4 days. 

Drug-induced liver injury (DILI) occurs as an unpredicted response to therapeutic doses of a medication (an idiosyncratic reaction–Type B) or as a response to a high dose of the medication (intrinsic toxicity–Type A), which are associated with hepatic injury [5,6]. Type A reactions are a direct result of the drugs’ therapeutic effect. The reaction is dose- and frequency-dependent. Type B reactions are unexpected, unrelated to the dose and frequency of administration, and only occur in a minority of patients who are presumed to have a predisposition [5,6]. 

In this paper, we present an example of denosumab-induced immune hepatitis. 

DILI encompasses both acute and/or chronic hepatic lesions. The liver injury may be the only clinical manifestation of the adverse drug effect, or it may be accompanied by injury to other organs or by systemic manifestations.

This article supports the safe and effective use of drugs by patients and guides laboratory medicine professionals in determining the possible DILI. We report a second rare case of DILI resulting from exposure to denosumab. 

## 2. Case Report

A 43-year-old female was referred to our Endocrinology, Diabetes & Metabolism Department for treatment of low bone mineral density (BMD) following endocrine therapy of breast cancer. She developed premature menopause at the age of 36 years, when she underwent a left lumpectomy due to an infiltrating duct carcinoma of the breast (T1 NO MO). She subsequently underwent adjuvant chemo-radiotherapy. The tumor was estrogen receptor (ER) positive. Due to her history of anticardiolipin syndrome, tamoxifen, a selective ER modulator treatment with an increased risk of thrombosis was contraindicated. She commenced treatment with a combination of an aromatase inhibitor (AI), letrozole, and a gonadotropin-releasing hormone (Gn RH agonist), lucrin, for estrogen suppression. The tablet contains 2.5 mg of letrozole, a nonsteroidal aromatase inhibitor. Femara [4,4′-(1H-1,2,4-Triazol-1-ylmethylene)] inhibits estrogen synthesis. There are no reported drug interactions between lucrin, letrazole or denosumab. 

The dual-energy X-ray absorptiometry scan T score in the lumbar spine was −2.2, and an oral bisphosphonate was started. In the next year, during bisphosphonate treatment, she developed gastrointestinal symptoms resulting in her treatment being changed to subcutaneous denosumab 60 mg every six months. She also took vitamin D and a calcium replacement. There was no past history of fracture. The patient did not smoke or consume alcohol. The only other therapeutic she took was aspirin.

On the third year after starting on denosumab and one month after she had received a subcutaneous administration of the drug, a gradual rise of liver enzymes was noted. The patient became tired and reported a poor appetite. The physical examination was unremarkable. 

Laboratory investigations revealed a complete blood count (CBC) and international normalized ratio (INR) within normal limits. An elevation of serum transaminases (ALT and AST) to around 10 times ULN was observed. Routine virology tests for the determination of hepatitis virus B (HBV) and virus hepatitis C (HCV), as well as Epstein–Bar virus (EBV) and cytomegalovirus (CMV), were negative. The routine autoimmune serology values for smooth muscle antibody, mitochondrial antibody and anti-nuclear antibody (ANA) were negative. The value for soluble liver antigen-antibody (Ag Ab) was 2.3 (0.0–20.0 units), the value for anti proteinase was 3 < 0.2 (0.00–0.99 index), and the value for anti myeloperoxidase was < 0.2 (0.00–0.99 index). The liver kidney microsome Ab-(LKM, liver kidney microsome andibody) was also negative. There was no evidence of chronic hepatitis. The GGT level was 67 U/L (0–38 U/L). The serum gamma-globulin level was 1.72 g/dL (0.7–1.6 gr/dl). The total bilirubin (TB) and serum albumin levels were normal. An abdominal ultrasound was normal.

Scheme 1 shows the longitudinal transaminases data in this patient, from the time she started denosumab, during hospitalization and after stopping the treatment.

The levels of the γGT follow the same pattern. As shown in Scheme 2, the enzyme levels spike during the liver toxic episode and decline after the interruption of the therapy. The elevation of this enzyme points to a cholestatic process.

## 3. Pathology-Methods

Samples in formalin were embedded in the Pathology Institute into formalin-fixed paraffin-embedded tissue (FFPE) blocks. The blocks were sectioned by Leica EM2245 microtome into 3–5 µm slices, which were loaded on StarFrost microscope slides for H&E staining or on Leica X-tra Adhesive slides for immunohistochemistry (IHC) staining.

Hematoxylin and eosin (H&E) staining slides were loaded onto Sakura’s Tissue-Tek^©^ Prisma & Film for the automatic staining and covering of the slide.

The IHC staining protocol includes: slides’ incubation at 60 °C for 45 min, xylen incubation for removal of paraffin, hematoxilyn (PRC LTd., 17610, Harris, Jefferson, OR, USA) incubation for 10 min (two washes for 5 min each) incubation in 70% ethanol for washing, 10 min of incubation in Eosin (Pioneer Research Chemical Ltd., 18553, Colchester, UK) staining, two washings × 5 min incubation with 70% ethanol, 2 washings × 5 min incubation in 100% ETOH, three times × 5 min in Xylen and cover slipping with Xylen film. All procedures were done according to the company’s protocol & recommendations. To read the fibrin deposits, Masson trychrome staining was employed. 

The slides were loaded on Ultra Bench Ventana and stained according to the validated protocol with cytokeratin (CK)7 Antibody (DAKO, Clone: OVTL12/30, Dil: 1:100, UltraView, standard, 32 min Ab incubation). An additional slide was stained with MUM1 Ab (clone: MUM1p; Dil: 1:50; DBS; Ultra View standard, 37 °C, 42′ + Amplifier). 

Stains were performed with a Ventana Ultra Bench stain apparatus using an ultraView Universal DAB Detection Kit (Catalog Number: 760).

## 4. Histology

A liver biopsy was performed, which revealed a moderate to severe chronic inflammatory infiltrate.

Specifically, the liver core biopsy revealed a well-preserved liver architecture. The parenchyma had a few foci of inflammation. In the portal tracts, there was a moderate to severe infiltrate containing plasma cells, numerous positive CD3+ small T lymphocytes, and few C 20 positive B lymphocytes and eosinophils (Figure 1, Figure 2 and Figure 3). Hyperplasia and mild portal fibrosis were identified using Masson staining (Figure 4).

Immunostaining revealed a severe interface hepatitis with bile ductular proliferation on CK-7 staining (Figure 5). Immunohistochemistry (Figure 6) presents an infiltrate containing MUM-1 positive plasma cells in the portal tracts. The likely diagnosis was considered to be consistent with immune hepatitis due to DILI. 

## 5. Discussion

Osteoporosis. is characterized by low bone mass, microarchitectural disruption and skeletal fragility, resulting in decreased bone strength and increased risk of fracture. Endocrine therapy, an adjuvant treatment in the management of patients with hormone-sensitive breast cancer is associated with adverse effects on the musculoskeletal system and commonly increases the risk of osteoporosis and fractures. 

Angiogenesis and osteogenesis connect for the bone function. Blood vessels supply oxygen, nutrients and progenitor cells to the osteogenic environment. The functional vasculature system cooperates for physiological bone healing [7].

The identification of novel therapeutic targets to prevent the disease aimed at inhibiting bone resorption and increasing bone formation. In our patient Denosumab was prescribed to treat aromatase induced bone loss.

The combination of the clinical illness together with biochemical abnormalities, the temporal association and the lack of any other suspects remains highly suspicious for denosumab-induced toxic events. The calculated score of 6 (a probable drug-induced adverse reaction) was obtained using the Naranjo score [8]. A causality assessment via the updated RUCAM was done [9]. The updated RUCAM takes into account divergent laboratory analyses of liver injury and classifies it into two different subscales: the hepatocellular type of injury and the cholestatic or mixed type of injury. These types can be differentiated using the ratio R, calculated as the ALT/ALP activity measured at the time of liver injury and expressed as a multiple of ULN. In our case, the liver injury was consistent with hepatocellular and cholestatic injury, and the RUCAM score was 4, leading to a *possible* causality level of DILI from denosumab. It appears unlikely that a direct hepatotoxic effect would appear after three years of use, and thus the possibility of an immune-mediated drug-induced liver injury was considered more likely. Oral steroid treatment with budesonide at 9 mg/day was commenced. Over the course of the following months, there was a slow decline in the levels of the transaminases and the GGT. Currently the liver tests are normalized, and the patient remains in good health on maintenance therapy with steroids.

Denosumab is an effective antiresorptive drug; however, discontinuation of the drug can result in an accelerated bone turnover, rapid loss of bone mineral density and an increased rate of multiple vertebral fractures. However, since the hepatic illness may recur upon reintroduction of the offending drug, the rechallenge of denosumab was not considered ethically possible, especially in view of its prolonged half-life. After denosumab was stopped, treatment was switched to intravenous zolendronic acid at 5 mg/year. The last dual-energy X-ray absorptiometry scan T score was −2.1 at the lumbar spine (BMD 0.925g/cm^2^), −1.6 at the neck of the femur (BMD 0.817 g/cm^2^) and −0.6 at the forearm (BMD 0.638 g/cm^2^).

We describe a case of a patient following a lumpectomy because of a duct carcinoma of the breast. Since the tumor expressed a hormone receptor, she was started on endocrine therapy. Consistent with recommendations from several international expert groups [2,3,4,5,6,7,8,10] for the assessment of aromatase inhibitor-induced bone loss, an antiresorptive treatment with oral biphosphonate was started. During the follow-up visits, the bisphosphonate therapy was switched to denosumab. On the third year after starting on denosumab, a gradual rise of liver enzymes was noted. The abnormal liver testing and the result of the liver biopsy suggested a diagnosis of DILI. The temporal relation, the prolonged half-life of the drug and the absence of other causes implicate denosumab as the possible cause of hepatotoxicity, so the drug was stopped. We believe that the liver damage was immune-related due to denosumab. The prolonged steroid treatment needed for the liver disease adds another factor with deleterious effects on the musculoskeletal system and increases the risk of osteoporosis and fractures in our patient. For the preservation of bone health, multiple risk factors are taken into consideration. The treatment varies from basic bone protective measures to specific antiresorptive therapy, and the treatment duration should last for as long as the course of the endocrine therapy is given [11]. For this reason, and because the rechallenge of denosumab was not considered ethically possible, it was extremely important to demonstrate the causality of the drug. 

Since the liver has a dual blood supply, both systemic and portal, many drugs can cause hepatotoxicity, including those affecting the musculo-skeletal system [12]. 

Menopause results in a decrease in estrogen concentrations and consequently an enhanced osteoclast maturation and activity. In addition, there is a decrease in the osteoblast function. RANKL and the osteoprotegerin (OPG) system have central roles in the the osteoclast activity,

RANKL is expressed on both preosteoblast and stromatolite cell surfaces and plays a central role in osteoclastogenesis by binding to the RANK receptor. RANKL plays an important role in osteoclast differentiation and activation and also in the inhibition of osteoclast apoptosis. OPG, secreted by both osteoblasts and stromatolite cells in the bone marrow, is a soluble member of the tumor necrosis factor superfamily that decreases the RANKL activity and thus inhibits osteoclasts’ differentiation, resulting in a decrease in the osteoclast activity [11].

Since Prolia was approved by the FDA in 2010 for the treatment of osteoporosis, it has been implicated in a large number of clinical trials. Denosumab was not associated with changes in serum aminotransferase levels, and the rates of adverse reactions were similar in patients who received denosumab or placebo. 

A study population has been published in 2006. The authors described 412 postmenopausal women with a low bone density treated with denosumab [three doses every three or six months], versus alendronate or placebo for 12 months. The bone density increased with denosumab and alendronate but not with placebo. The authors did not find changes in the blood chemistry results [4].

Lewieck et al. [13] followed up the 412 women treated with denosumab, alendronate or a placebo. The incidence of side-effects was similar in all three groups and did not change during the second year of the study. There were “no clinically relevant changes in either serum chemistry or hematology values” [13]. Another study of 255 women with breast cancer and bone metastases treated with denosumab for at least 13 weeks found no “unexpected” changes in liver enzymes [14]. Cummings et al. (2009) reported a randomized placebo-controlled trial of 7868 postmenopausal women with osteoporosis treated with denosumab or a placebo [every six months] for three years. Bone minerals increased and there were fewer fractures in the denosumab-treated patients. The adverse events were similar except for eczema [3% vs. 1.7%], cellulitis [0.3% vs. <0.1%] and flatulence [2.2% vs. 1.4%]. The authors did not describe ALT elevations or hepatotoxicity [15]. Bone et al. (2013) followed 3547 postmenopausal women with osteoporosis treated with denosumab for three or six years. The authors declared that the side-effects did not increase with time; they did not note transaminase elevations or DILI [16]. An additional study described no hepatotoxicity in 4550 postmenopausal women with osteoporosis that were treated for another seven years with denosumab. Serious adverse events included osteonecrosis of the jaw (*n* = 13) and atypical fractures, but there was no mention of ALT elevations. In their systematic review and meta-analysis where the authors described the efficacy and safety of denosumab in the therapy of bone metastases, they did not mention ALT elevations or hepatotoxicity [17].

Moreover, the Drug Induced Liver Injury Network (DILIN) group reported that among 899 cases of drug-induced liver injury enrolled in a US prospective study between 2004 and 2013, no cases were attributed to denosumab or other agents used to treat osteoporosis [18].

The only reported case of DILI related to Prolia has been reported by our group at Kaplan Medical Center [7]. In this case, to identify the correlation between denosumab and the patient’s liver toxicity, the authors examined the temporal relationship, the amount of drug administered and the duration of the treatment with denosumab. Exposure to over-the-counter medications and both herbal and dietary supplements was ruled out. A lymphocyte toxicity assay (LTA) was performed to determine if denosumab was involved in the liver injury. LTA is based upon incubating the lymphocytes of the specific individual with the drug that produced the reaction (Denosumab) in the presence or absence of a cytochrome P450 system. After 24 h, the lymphocytes are exposed to the yellow tetrazolium dye 3-(4,5-dimethylthiazol-2-yl)-2,5-diphenyltetrazolium bromide (MTT). The LTA of the patient indicated 31% toxicity upon incubation with the monoclonal antibody vs. control. This result implicated denosumab possibly being responsible for the liver reaction. Primary biliary cholangitis and other chronic liver diseases are associated with osteoporosis. There are reports showing lower levels of RANKL in these diseases than in controls [19]. More than this, circulating mononuclear cells could have a higher capacity to differentiate into osteoclasts in patients with chronic liver disease and osteopenia [20]. 

Presently, Osteoporosis Canada Clinical Practice Guidelines recommend this drug as a first-line therapy for postmenopausal osteoporosis [21]. Wensel’s group summarized serious denosumab-induced infections, dermatologic adverse reactions, as well as possible hypocalcemia [22]. Adverse effects to denosumab are reported in dentistry. A 74-year-old woman treated with denosumab developed osteonecrosis of the mandible [23]. In a recent review article [24], the authors studied bone biopsies collected from patients exposed to denosumab that presented infectious, inflammatory and necrotic jaw diseases. They found different degrees of bone histological changes. However, none of the individuals presented hepatic disturbances [24].

In our first description of denosumab-induced DILI [7], we demonstrated a possible mechanism by which denosumab could cause severe liver toxicity: by blocking and actively reducing RANKL under normal blood levels. In the first report, we described a 72-year-old woman. The liver biopsy revealed a sub-massive hepatic necrosis consistent with DILI. It was suggested that denosumab might produce a cholestatic reaction. The drug also induced the elevation of members of the TNF family and inflammatory chemokines. Because of the prolonged half-life of denosumab (25.4 days), this immune activation can be a long-lasting event. In the present case, the person is also a woman, but she is younger.

It is not clear why the onset of the DILI occurred so long after the initiation of Prolia therapy. This suggests immunological priming and, together with the response to steroids, is consistent with an immune mechanism and not autoimmune hepatitis. There is a single report of the production of denosumab-binding antibodies. Two patients who were treated with 100 mg of denosumab administered every six months developed such antibodies, after a month of treatment in one patient and after 12 months in the other. Upon a bioassay evaluation, these were not found to be neutralizing antibodies.

They were not detected in subsequent samples [4]. The long half-life of denosumab may play a role in the development of DILI, with a prolonged exposure at low doses. Denosumab reported worldwide sales of $701 million in the third quarter of 2020.

It is imperative to follow up on patients for a longer period of time in order to ensure the safety of this drug.

## Data Availability

Data supporting reported results can be found at the hospital recording database.

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
