# Peer review of "Denosumab-Induced Immune Hepatitis"

_biomedicines, 2021, doi:10.3390/biomedicines9010076_

Round 1
Reviewer 1 Report
Since liver damage caused by Denosumab is extremely rare, it is described with carefully that DILI is caused by Denosumab. The author is also studying that can be done using patient samples.
Please improve the following one point.
Discussion; Since the chapter division of discussion is not clear, it gives a strong impression of redundancy. Please translate the chapter to clarify the points of the topic.
Author Response
Thank you very much for reviewing our paper. We appreciate tremendously your expertise.
We took in consideration your suggestion and omitted the redundancy in the discussion. Also we made additional changes in the description and kinetics of the case.
I hope that you will consider the new manuscript more interesting.
Reviewer 2 Report
Dr Ostrovsy et al report a case with Denosumab-induced immune hepatitis. This is the second case of hepatotoxicity resulting from Denosumab. A 43-year-old female was diagnosed as breast cancer and was treated with letrozole and lucrin. Denosumab was started to prevent osteoporosis. On the third year, hepatitis developed. Biopsy showed an immune-mediated process.
Comments:
- Case reports are important references to clinicians during their practice, albeit less frequently being cited. Personally, I encourage case report publication.
- In the title, please indicate in any way that this is a case report, not to confuse the readers with the concept that this is a general effect.
- If bilirubin and albumin and prothrombin time are all normal during the course, the authors should not use the term “liver failure”, as there is no functional impairment. Please just say “hepatitis flare”.
- Please provide the results of ANA, anti-smooth muscle antibody, anti-mitochondria antibody data, or other markers related to autoimmune hepatitis.
- Please provide a figure showing the clinical course with longitudinal transaminases data together with other relevant information including key drugs used in this patient, and the time to start and to stop these treatments.
Author Response
Thank to the reviewers we have been able to make the manuscript more focussed. As requested we made the following changes:
1- We change the title of the article
2- We deleted the term "liver failure"
3- We introduce more data to describe the kinetics of the transaminases and GGT from the beginning of the therapy continuing to the period of hospitalization and the stop of therapeutic agent.
We appreciate very much your constructive effort to imptove our paper.